# Attentional Neurodiversity in Physical Education Lessons: A Sustainable and Inclusive Challenge for Teachers

**Miguel Villa-de Gregorio** [1,*], **Miriam Palomo-Nieto** [1], **Miguel Ángel Gómez-Ruano** [2] **and Luis Miguel Ruiz-Pérez** [2]

1   Facultad de Educación-Centro de Formación del Profesorado, Universidad Complutense de Madrid, 28040 Madrid, Spain
2   Facultad de Ciencias de la Actividad Física y del Deporte (INEF), Universidad Politécnica de Madrid, 28040 Madrid, Spain
*   Correspondence: mivill03@ucm.es

**Abstract:** Attentional neurodiversity is evidenced in the majority of current schools. The role that physical education plays for the inclusion of students with attentional problems in the school is quite relevant. This essay aims to show the effectiveness of sports and physical exercise on the core symptoms of schoolchildren and adolescents with attention deficit/hyperactivity disorder (ADHD). What is more, this proposal sheds light the key differences between clinical conditions of physical exercise and/or sport interventions and the ecological conditions of physical education lessons where the students with attentional problems should be included. Finally, this essay puts forward the real need for bridging the gap between physical education and science by changing the curriculum based programs, re-designing the teachers' training programs, and acquiring the scientific recommendations in order to ensure the inclusion of all students according to Agenda 2030.

**Keywords:** physical education; sport; inclusive education; special needs education; sustainable development

## 1. Introduction

After 15 years of unprecedented progress in achieving the Millennium Development Goals (MDGs), the world has turned its attention to its successors, the Sustainable Development Goals (SDGs), in a period of transition towards the Agenda 2030 for Sustainable Development that was recently approved [1]. In reviewing the achievements and unfinished business around the eight MDGs, the international community, led by the United Nations, undertook a comprehensive consultation process with stakeholders from all spheres of society and agreed on the 17 SDGs to be pursued in the next 15 years [2]. With the overarching aspiration to bring people and the planet closer together and leave no one behind, the 2030 Agenda represents a unique opportunity to inspire global action for development around the world [2]. Education has been proposed as a fundamental axis for achieving the SDGs; thus, all the components of the education system are essential for contributing to a more sustainable world at an economic, social, and environmental level, and to achieve dynamic and sustainable development through a general framework of action whose motto, "Leave no one behind", is directly rooted in the fundamental objectives of inclusive education [1,3]. When the 2030 Agenda was approved, international organizations agreed to recognize that education is an essential means for the achievement success of their set goals [4]. It is in Sustainable Development Goal 4 where this intention is reflected with all clarity, since it aims to "Guarantee an inclusive and equitable education quality and promote lifelong learning opportunities for all" [1].

The movement of educational inclusion, which emerged in the 1990s of the last century, refers to an endless search for the best way to address the need for diversity [5]. It is based on the principle of equity, to respond to each student's personal and social needs of each

student by helping them to overcome any problem of learning [5]. It also appeals to social justice through the implementation of activities that make it possible to acquire a basic level of skills, which requires new teaching–learning approaches that provide the school success of all students through an equitable and quality education [6]. Therefore, inclusive education must respond to all students' personal, psychological, and social characteristics [7]. Likewise, inclusive education should be part of the integrating school where the teaching process involves all the children, regardless of their educational needs.

This new perspective tries to avoid the deficit model, focused on assimilating inclusive physical education (PE) with the response for students with special educational needs or, according to what is established in the Spanish curriculum, for students with educational support needs, which so often determines parallel educational responses in centers that favor their exclusion more than its inclusion [8].

Furthermore, the paradigm of inclusive PE aims to ensure that PE reaches everyone by providing the means for teachers to give educational responses that guarantee equality and social justice by educating diversity successfully [9].

To promote the students' social and academic life, we should take the teaching plan into consideration [10]. Therefore, PE teachers must provide the students' autonomy and self-regulation of learning by giving priority to a common curriculum and teaching cooperative learning structures [9]. Thus, each action of change must be established according to the characteristics of each school, its context, teachers, students, and community involvement [4]. The education offered by the schools must guarantee the inclusion of all the students, being essential both the role that each school assumes in this change as the support of the political, educational, and social instances that are received for the development of improvement actions that go beyond purely school action to develop social and civic actions [6].

We must be aware that, although the specific measures to serve the needs of diversity can be useful in schools that are inclusive in terms of the extra resources provided by itself, it must also be considered that focusing the attention on the most vulnerable students through these measures can create parallel paths within schools, thereby contributing to the exclusion of them. Then, the acceptance of differences and absolute respect for the innate heterogeneity between people is an essential step to laying solid foundations for a quality school for all to achieve SDG 4 [7].

"To include" is to think about all the diversity of students, offering a varied and differentiated educational response to students with very different characteristics in terms of ways of learning, being, and relating [10]. From this perspective, the diversity of all people is recognized and accepted without trying to match them or change their characteristics, focusing attention on changing the context and developing strategies to respond to this diversity. Since all the students are diverse, it is necessary to promote learning for all from belonging to the group and respect for individual characteristics [10]. This position leads to an inclusive school model, which is characterized by serving all students and responding to the real and natural diversity of any human group [7,10].

The term "inclusive physical education" implies, therefore, an assessment of the heterogeneous characteristics of boys and girls as an enriching element and as a possibility of achieving more optimal educational results [8,9]. Therefore, PE teachers must offer their students the widest possible variety of experiences and activities. Likewise, a quality inclusive PE ensures, for all students included in it, access to learning and knowledge in the most optimal conditions possible [9,10].

Finally, under this paradigm, the PE teacher's management of a diverse classroom in which students with ADHD are enrolled represents a great challenge. Because of the growing presence of ADHD among children and adolescents in schools, as a main part of attentional neurodiversity, this essay aims to highlight how important is to bridge a gap between science and PE in order to respond to the educational needs of them.

## 2. The Needs of Attentional Neurodiversity

*A Special Psychosocial Profile Included into a Sustainable Physical Education Framework*

Attentional deficit/hyperactivity disorder (ADHD) is a common disorder that, although it is most frequently diagnosed during school years, affects individuals across their lifespans [11]. It is associated with a wide range of psychiatric difficulties [12]. Furthermore, children with ADHD are typically characterized by developmentally inappropriate levels of attention, hyperactivity, and impulsivity that cause significant impairment in daily activities, such as excessive motor activity, difficulty taking turns in class, excessive talking, interrupting others, etc. [13]. Other researchers [14] also considered that children with ADHD may show emotional disorders. Likewise, these investigations found that, among Spanish preschoolers, the prevalence of this disorder was 5.4%, lower than in the study of Catalá-López et al. [15], where these researchers found a 6.8% prevalence in Spanish children and adolescents.

Additionally, there are three subtypes of ADHD: (i) Combined: manifests three symptoms—hyperactivity, inattention, and impulsivity; (ii) The predominance of attention deficit: the symptom of which is inattention; and (iii) Hyperactive–impulsive predominance: hyperactivity and impulsivity behavior predominates [16]. The hyperactive-impulsive and combined types are the most prevalent ones. Furthermore, these types are related to some psychosocial problems, such as participation and social preferences for learning in school [17].

On the other hand, ADHD and developmental coordination disorder (DCD) are both childhood disorders identified in the DSM-5 [18] with a very similar population prevalence [12,19]. Thus, it is widely accepted that ADHD is a co-morbid disorder. Likewise, children with ADHD often demonstrate poor motor coordination or motor performance and balance [20]. Clinical and epidemiological studies report that 30% to 50% of children with attentional problems suffer from motor coordination problems [20]. Therefore, this reality could be extrapolated into PE lessons. As researchers, we should wonder how PE contributes to the inclusion of students with ADHD.

Apart from the healthy benefits of PE, it is a transcendental subject that could contribute to creating a context that allows the development of cooperation, friendship, and fellowship [20]. These virtues allow children with ADHD to improve their adaptation to norms, social skills, self-esteem, care, and feelings of support that are significantly related to the improvement of their participation and social preferences for learning in PE classes [21,22].

In daily school life, students with ADHD manifest behaviors that the literature has described as disruptive while performing tasks [23]. Furthermore, during periods of free play, these boys and girls perform fewer activities and participate in more solitary games. Moreover, they usually manifest some negative behaviors, such as aggression and provocative social responses. In short, they show serious problems establishing friendships with other children; that is why later behavioral and emotional imbalances could occur [23]. To understand why students with ADHD usually have some social problems in their daily school life, researchers first refer to social devaluation, arguing that students with typically developing (TD) sometimes exclude those peers whom they perceive as different from themselves [23]. Second, excluding behaviors are highlighted by researchers, because students with TD can use derogatory comments towards those boys and girls with whom they have no affinity [23]. Finally, researchers differentiate the reputation bias, which refers to the adoption of a negative perspective through which students with ADHD are valued and which is difficult to modify when the pattern is consistently established [23].

Under the premise of inclusion, within the school environment, it is recommended to create and structure situations in which peers are involved [23]. PE teachers are responsible for creating the right pedagogical conditions to obtain positive outcomes. It is the responsibility of PE teachers to create pedagogical circumstances under which positive outcomes can be obtained. Some strategies, such as involvement in structured cooperative interactions within the classroom, training peers in social skills, and, finally, redirecting

attitudes of rejection of equals towards attitudes of acceptance of differences, could be interesting [9,24]. Through this kind of strategy, PE teachers guide students in adopting an inclusive and welcoming perspective toward their peers with ADHD. To deal with exclusionary behaviors among peers, PE teachers encourage compliance with a series of clear rules in class, thus contributing to social inclusion and the organization of activities that facilitate the formation of positive social ties [25] To address this issue, during PE lessons, cooperative learning, in particular, has a broad base of support among physical educators who have documented increased motivation [25]. Likewise, there is some evidence that students with ADHD, who were in PE lessons with cooperative learning, found more friendships [26,27]. Social benefits are generally student perceptions or attitudes, such as social cohesion, acceptance, desirability as a work partner, and task-related behavior, such as giving and receiving help and task resources [28]. It is well known that the majority of PE curricula are based on cooperative learning by applying a wide range of sports and other content that promotes this kind of learning. Moreover, it would be important to talk about fellowship as a social preference for learning in PE because it helps to develop teamwork and cooperation among students [29] apart from making group decisions and working towards common goals [30]. What is more, personal and social development constitutes one of the main and most frequently cited goals of European PE programs [31] The real fact is that PE as a subject contributes in a natural way to develop some prosocial behaviors, such as respect, empathy, and sympathy. Cooperation and work ethic are improved, as well, in the context of PE, by developing help for peers and teamwork. [31]. Likewise, the literature expressed an increase in goal-setting, responsibility, leadership, meeting people, and making friends, as well as communication among students through PE [31] Concerning students' participation in PE studies, it was revealed that, through PE, children learn to take turns, display empathy and respect, and are able to handle or deal with conflicts [31] Furthermore, some studies showed that PE had a positive effect on self-control, coping skills, problem-solving skills, and assertiveness, although it is evident that there is a lack of specific information regarding the program characteristics [32–34].

Once it has been shown in the scientific literature that there is social inclusion of students in PE lessons, regardless of having ADHD or TD, it would be convenient to know how far PE is from science with regards to achieving the full inclusion of students with ADHD.

## 3. Physical Exercise and Sport as Means of Alleviating the Effects of Attention Deficit/Hyperactivity Disorder

Nowadays, there is a wide range of treatment methods for ADHD, such as drugs and behavioral or psychological therapy [35]. Related to its causes, currently, there are some ADHD hypotheses based on epigenetics factors [35]. Notwithstanding, the theory based on the dysfunction of the catecholamines (norepinephrine, dopamine, and epinephrine), responsible for some mechanisms of attention, concentration, and arousal state, is one of the most known [36]. This is why drugs, such as methylphenidate or amphetamine, are currently the mainline in the treatment for ADHD because they can attenuate the symptoms of ADHD among schoolchildren and/or adolescents [37,38]. However, there is a huge number of scientific studies that report the effectiveness of exercise and/or sports to enhance the release of some catecholamines in the prefrontal cortex among other brain parts [37–39]. In addition, research indicates that exercise could provide similar benefits to those provided by medication and suggests that the physiologic benefits of exercise are similar to those of pharmacological interventions, specifically in terms of possible increased catecholamine response [40].

Down below, this part of the present essay aims to analyze the effects of structured physical activity and or exercise forms designed as an intervention program on schoolchildren and adolescents diagnosed with ADHD.

First of all, scientific studies found that regular physical activities could provoke a lot of benefits in physiological and psychological terms; the improvement of motor coordination,

cognitive mechanisms, physical fitness, and interpersonal skills are some examples of multiple advantages of applying a planned and structured physical activity on children and/or adolescents [41,42]. However, the current research poses the following question. How much is exercise intensity important to reduce the symptoms of ADHD among children? Researchers should be aware of the importance of designing and applying a physical activity or exercise intervention program based on moderate-high aerobic drills. In fact, when the intensity of physical activity is not too high, it takes the place of an increase in brain-derived neurotrophic factor (BDNF) due to the presence of peripheral lactate levels in the brain, thereby activating the central nervous system, as well as enhancing the information processing capacity by improving the amount of dopamine in the brain [43,44]. Furthermore, moderate intensity exercise could decrease impulsive behavior and inattention in children and adolescents with ADHD [37,38,45].

Secondly, it would be interesting to highlight the relationship between the intensity of exercise and executive functions. Likewise, we should know that executive functions are a category of cognitive dimension and refer to the group of complex cognitive abilities that control the skills required for goal-directed behavior [46,47]. Working memory or impulse inhibition, for example, are responsible for organizing tasks, managing time, or solving problems. Thus, we are wondering about the importance of physical activity and exercise to improve this kind of cognitive function.

Some studies found that children with a high level of daily physical activity had better working memory and inhibitory control performance because of their increasing aerobic endurance and fitness [48,49].

The scientific literature reviewed showed that, when the exercise intensity was high, ADHD symptoms, such as inattention and impulsivity, decreased; thus, some cognitive functions could increase [50,51]. These studies highlighted the application of high-intensity interval training (HIIT) as an effective way to reduce ADHD symptoms. Moreover, apart from reducing the core ADHD symptoms, long periods of HIIT also improved fitness, motor competence, and other psychosocial aspects, such as self-confidence or behavioral symptoms [42]. Therefore, taking the main characteristics of children with ADHD (impulsiveness, impatience, lack of motivation, and persistence) into consideration, interval training is a strongly recommended form of exercise to reduce ADHD symptoms. What is more, a huge number of pieces of evidence indicate that aerobic and anaerobic exercise and sports could provoke improvements in working memory, inhibitory control, cognitive flexibility, and attentional processes among children with ADHD [52–59].

It is complicated to set a motivational environment to engage children with ADHD to practice physical activity or do any sport; that is the main reason why the content of exercise programs or interventions should be based on games [37]. There is a study in which 37 children with ADHD underwent eight weeks of an exercise program based on games. The results showed a significant increase in terms of complex information processing capacity [60]. Other studies found that planned quantitation and qualitative exercise have positive effects on attention and executive function [61–63], which developed during during sports game intervention for eight weeks among children with ADHD (three sessions of 30 min), and they achieve an interesting result in terms of executive function improvements. Hattabi et al. [64] discovered water aerobics games as an appropriate tool to reduce some symptoms of ADHD among children, such as inattention and hyperactivity. Similarly, Silva et al. [61] found that swimming is an effective treatment to reduce some symptoms of ADHD.

What is more, a growing number of studies provide information that strengthens sensory–motor processes, such as sensory stimulation, regulation of muscle tension changes, visuomotor coordination, and vestibular sensation and proprioception, which contribute to improving attention performance in children with ADHD [35]. Dancing with some balance, proprioceptive, and rhythmic elements contribute to improving the attention of children with ADHD [65]. In addition, other types of exercise or physical activity programs or interventions are based on mindfulness and meditation. Some studies analyzed the effect

of yoga and meditation on children with ADHD and found that their self-control ability and attention awareness are highly improved [66].

In conclusion, physical activity, exercise, and sports are very important to treat and reduce some symptoms of children and adolescents with ADHD. Aerobic and anaerobic drills, sports, and games are strongly recommended, and they should be administered in curricular and extracurricular contexts. Thus, structured exercises should be considered an essential treatment for reducing the symptoms of children with ADHD in the physical education (PE) classroom. However, the referenced studies that have been reviewed in the present essay were not curriculum-based PE programs and did not report dosage and/or progression parameters. Particularly, children with ADHD would be benefitted from a school-based, structured exercise program/intervention, which can be incorporated within the PE curriculum [67].

## 4. How Physical Education Is Far Away from the Structured Exercise Programs: The Way toward a Sustainable and Inclusive Scenario for Schoolchildren and Adolescents with ADHD

It is well known that the effects of exercise at moderate to high levels of intensity lead some improvements in cognitive function, fitness, motor skills, and behavior, regardless of children with or without ADHD. However, the majority of studies are under experimental conditions with a huge difference in their methodologies. This kind of program does not implement many adaptations that could increase participation and enjoyment among schoolchildren with ADHD [67]. Likewise, there should be applied ways to make the physical education environment more conducive for students with ADHD and to enhance learning for all students. Moreover, the structured and school-based exercise programs do not keep the specific needs and contexts of the PE lessons in mind. It should be considered that all schoolchildren are different, even those with ADHD. Thus, the specific exercise and its benefits for children with ADHD in physical education lessons remain unclear. Furthermore, PE teachers use a wide range of strategies aimed to be generally inclusive, rather than specifically designed interventions for ADHD children alone [67]. Thereby, specific activities or adjustments to the PE lessons are not included.

Physical education is currently in the majority curriculum of the schools and therefore is a proper location to include a structured exercise program to manage schoolchildren with ADHD. This kind of program would contribute to improving not only their fitness, but also some psychosocial parameters [27]. The structured and school-based exercise programs do not consider many adaptations related to equipment management, class organization, cooperative learning, peer tutoring, classroom management, discipline and rewards, routines, etc. [68]. Likewise, physical education lessons are developed in an educational context different from the ordinary classroom. Considering the contextual characteristics of PE and the situations that may arise among the students, it is imperative to know and control methodological aspects of PE teaching, as well as the educational curriculum by which it is governed. For this reason, it is observed that the physical activity programs designed for the improvement of symptoms of schoolchildren and adolescents with ADHD could be far from the reality of physical education classes. Thus, a series of arguments are presented below to describe the scenario in which a student with ADHD develops during PE lessons.

### 4.1. Constraints to Learning in Physical Education

In physical education lessons, schoolchildren with ADHD are usually exposed to a huge number of distractions due to there being many more classmates, and the organization and structure in the sports facilities are more complicated than in a regular classroom [67]. The standard PE lessons in school can be challenging for children with ADHD due to long periods of the same activity, or periods of inactivity in team sports. That is likely to cause significant impairment in daily PE lessons, such as difficulty taking turns, impulsivity, excessive talking, interrupting their peers, etc. [27]. Furthermore, PE teachers know little or

nothing about how to set the adaptations they can make in the PE lessons to create a better teaching–learning environment for their pupils with ADHD [69].

### 4.2. Social Preferences for Learning in Physical Education

During PE lessons, there is a wide range of situations in which schoolchildren and adolescents work as a team, stimulating the learning of their peers to achieve common goals; these are situations in which every student assumes his/her role individually. In this respect, it is worth mentioning that the scientific literature analyzed does not show any curriculum-based PE program and its effects on the social preferences for the participation of students with ADHD during PE lessons.

However, in Spain, some researchers examined the effect of a 12-week curriculum-based PE program on social preferences for learning in PE in Spanish secondary students with ADHD and typical development (TD). The sample consisted of 13 students with ADHD (nine boys and four girls, aged 15 years) and 13 students without ADHD (nine boys and four girls, 15 years old). Before and after the PE program, all participants completed the Graupera/Ruiz Scale of Social Interaction Preferences in PE Learning [70], which analyzes four learning preference dimensions: cooperation, competition, fellowship, and individualism. After the PE program, the students with ADHD showed an increase in their cooperation, competition, and individualism scores. They did not show a significant increase in the fellowship dimension [27].

### 4.3. Achievement Motivation in Physical Education

The motivation to learn in PE lessons must be a factor to take into account when studying and evaluating the benefits of the aforementioned curricular area for students with ADHD. Similarly, the intervention programs exposed throughout this essay do not take into account some psychosocial characteristics of schoolchildren with ADHD, such as the fear of failing in front of their classmates, probably conditioned by the low levels of comparative motor competence, little commitment to the PE area, probably derived from low perceived self-competence, or the feeling of belonging to a group [71].

The same researchers [71] designed a study to analyze the effect of 12-week curriculum-based physical education lessons on achievement motivation, after 12 weeks, in a group of schoolchildren with ADHD compare to another group that is typically developing (TD). The results showed that schoolchildren with ADHD were less committed to learning in PE classes and had a lower perception of their motor competence compared to their peers with TD. School children with ADHD, in both assessments, showed higher levels of anxiety when making mistakes than their peers with TD.

### 4.4. The Curricular Context of Physical Education and the Context of Interventions Programs

The intervention programs exposed throughout this essay use programmed and structured physical exercise around variables of time, intensity, and density, which do not correspond to the curricular reality. In Spain, the teaching load in the PE area is two weekly sessions of 55 min. Usually, the teacher–student ratio is usually very high, which makes it difficult to implement tasks, such as those that appear in some of the analyzed programs.

Furthermore, in terms of sports facilities, there is a huge difference between structured exercise programs and PE lessons. Thereby, Durgut et al. [72] used treadmills and vibrating platforms as means for the development of their intervention program. Likewise, Benzing and Schmidt [63] used video games as a means of developing their interventions.

What is more, some studies took place in extracurricular contexts instead of during PE lessons. Kadiri et al. [54] used martial arts in a private gymnasium. In the same way, other extracurricular programs were developed, such as the studies published by Silva et al. [65], developed in aquatic facilities and in which they used hippotherapy.

### 4.5. Other Differences between Physical Education and Interventions Programs

Finally, PE teaching is characterized by different pedagogical models: perceptual-motor, physical conditioning, personal and social development, etc. In addition, learning environments are another aspect to take into consideration to encourage the development of self-esteem, autonomy, cooperation, and tolerance, based on fair play among students with ADHD [34]. Otherwise, any structured exercise program shown throughout this essay was not taken into account. Likewise, the different structured exercise programs did not describe any characteristic related to [32].

1. Teaching styles: PE teachers should try to strike a balance between traditional instruction and the use of styles that promote active student participation to turn PE spaces into friendly environments and the courses into learning experiences.

2. Motor commitment time in PE: PE teachers should avoid students being stationary for long periods and waiting their turn to practice, thus increasing the amount of time that the majority of students were physically active.

3. Feedback is given: the PE teacher usually circulates close to where students are working to offer feedback and demonstration when necessary.

4. Individualization: every student should work at their level at an intensity level felt comfortable.

5. Improvement and equal opportunities: the PE teacher also gives students a wide range of opportunities for individual and small group practice through thoughtful use of equipment.

In conclusion, PE lessons are designed to promote high levels of physical activity that will improve health-related fitness, promote movement skills that add to success and enjoyment in physical activity, and encourage positive socialization regardless the students having been diagnosed with ADHD or not [32–34]. This may be part of an inclusive pedagogical approach, as compared to an additional needs-based approach to inclusion.

### 4.6. What about the Training of Future Physical Education Teachers? The Capability to Meet the Needs of Diversity

The discourse of inclusive education must guide educational and social policies, in addition to being present in schools and classrooms, and it must not be reduced to principles and ideals that do not transform educational reality. Thus, we must continue working on the advances in terms of the curriculum framework, curriculum goals, and methodology to achieve the goals of Agenda 2030 [73].

In the short and long term, the initial and continuous training of teachers should be set as an objective. On the contrary, with incapable and uncommitted teachers to inclusive education, SDG number 4 will not be possible [74].

Specific training directed to specialists (therapeutic pedagogy or auditory and language teacher specialists) does not solve the problem. Therefore, it requires specific training for all teachers to contribute to a real pedagogical renovation around the inclusive model along the stages of the educational system [73].

The teacher training diploma should qualify PE teachers to serve all students, without exception. However, it is well known that the Spanish University does not provide enough specific knowledge to future PE teachers. Thus, they have to obtain a Master's degree to acquire the knowledge that lets them become qualified to relate to the diversity in mainstream schools. In this sense, professors should provide students with education according to the needs of current society to form capable future teachers. This implies that, apart from the knowledge of the professors, the university curriculum should be reviewed and changed to be oriented toward sustainability. Thus, it should be based on the theoretical and practical bases of inclusive education to be able to serve the characteristics of diversity present in classrooms (gender, capacity, culture, etc.) [73].

The Spanish Master's degree in Teacher Training Compulsory Secondary Education and Baccalaureate is changing little by little, but it still requires improvements that make it possible for capable teachers who know the foundations of inclusive education. Nev-

ertheless, in the secondary education stage, the curricular and other types of gaps are more evident. The training will be structured around a specialized curriculum by areas of knowledge in which practices are key elements for consolidating the contents referring to the differences in individuals and sociocultural factors, which will result in the formation of reflective and critical teachers regarding their practice, and they will possess the necessary skills to adapt to teaching students that are more vulnerable, which will facilitate their learning [73,74].

Focusing attention on children with ADHD in mainstream schools, it is evident that their presence has increased in recent years [75]. Therefore, PE teachers must know the difficulties presented by students with these characteristics, not only as a form of observation or detection, but also as training to apply a correct and quality educational intervention [23]. Thus, in this ADHD and physical education binomial, there is a figure that plays a role determinant in the process of inclusion of students with this disorder in the PE classroom: the PE teacher.

This highlights the need to know, in depth, about the pathologies, disorders, or any type of problem that this kind of student may have. This need for better training in the healthcare field to diversity by specialist teachers in physical education was reflected in a study by Rodríguez et al. [76]. Likewise, another study showed the need to set a university curriculum made up of specific contents by teaching them through continuous training and professional programs, as well as recycling or self-training to know the wide range of disorders among schoolchildren and adolescents in mainstream schools [76].

Peña and Peña [76] showed that the knowledge of PE teachers about the wide range of neurodiversity disorders, such as ADHD, was not enough. They were not provided with a qualified training program during their Bachelor's degree. Likewise, not only were they reluctant to choose some specialized subjects in neurodiversity disorders or special education, but they also lacked continuous training and professional recycling.

The results showed the deficient academic training of PE teachers in the field of attention to diversity, particularly among PE teachers with broad professional experience. This lack of training is also reflected in another study when the researchers addressed the perception of PE teachers about their professional competence regarding inclusive education [76].

The inadequate training (the current curricula for future teachers in primary education, which improves the old diploma plans, offering very basic and general training in this sense) must be complemented with continuous training, as well as professional recycling oriented towards the knowledge about the presence of certain students with concrete disorders in the mainstream schools, such as ADHD [73]. This specificity in teaching and professional training would not only ensure confidence in themselves, but it would also be a positive reinforcement to face the hard task of facing the heterogeneity of groups, such as those found today in ordinary classrooms. Apart from the presence of better training for PE teachers in the field of attention to diversity before and during their professional careers, in many cases, a change is therefore needed by including a PE assistant able to serve all the students optimally, regardless of having ADHD or not [76].

In sum, the current university curriculum from the different Spanish Bachelor's degrees and Master's degrees (sports science degree and teaching diploma) does not guarantee adequate training to form qualified PE teachers for treating diversity, such as the attentional neurodiversity that they can find in PE classrooms. Therefore, the deficient curricula should be compensated with regulated training plans (promoted and organized by the competent educational authorities) and self-training plans or professional recycling.

### 4.7. Re-Thinking the Physical Education Lessons to Serve the Needs of Attentional Neurodiversity

The area of physical education, due to its intrinsic characteristics (motivation, playful environment, strengthening of social relations, teamwork, promotion of personal, social, and civic values) is shown to be a propitious environment to work with with regards to all students, regardless of having ADHD or TD, and it is important to contribute towards

the development of respect, cooperation, or coeducation, all items related to SDGs from AGENDA 2030 [23]. Likewise, for those students with ADHD, PE is a transcendental subject that could help to manage some difficulties in their daily school life, particularly during PE lessons [71]. Thus, PE teachers should be aware of the importance of rethinking the dynamic of their PE classes to serve attentional neurodiversity.

First of all, for schoolchildren with ADHD, the beginning of the PE lesson is essential, since the moment in which the relevant aspects of it are going to be presented to them is important [77]. If, when the students with ADHD arrive at the gymnasium, the equipment is messy and visible, or if the balls are spread over the court, they will likely want to start playing with them. Therefore, it seems logical that having everything organized may be a good way to keep the attention of this kind of schoolchildren [77].

The attention of schoolchildren with ADHD could be caught if the class is organized appropriately. They should be placed in the first rows to encourage them to be focused on what is transferred to them to practice and avoid being distracted by other classmates. It also favors eye contact, and the teacher should know them by name and notice when they disconnect from the task so that they can act accordingly [78].

When the class has finished, this closeness can be useful to highlight what has been practiced so that they can reflect on it and provoke a moment of greater tranquility.

Be that as it may, catching the attention of schoolchildren with or without problems is a key issue for the PE teacher.

What is more, PE teachers should be able to design and implement some drills, games, or motor activities based on cooperative learning. Cooperative learning promotes interaction among students with ADHD and their peers with TD, provoking great contact. Furthermore, cooperative learning improves social and individual responsibilities. Moreover, through cooperative learning games, PE teachers can promote equality and inclusive PE among the students because all the students playing in a cooperative group must value their teammates ´to work to achieve a common goal [70,71].

Secondly, PE teachers should be able to create a pedagogical context that creates a wide range of authentic sports experiences. Through this kind of experience, PE teachers would contribute to the development of teamwork, autonomy, and empathy among students with ADHD. As an example, PE teachers can offer students with ADHD different roles, such as coach, physical trainer, mediator, referee, etc., which could enhance their social skills [27,71].

Likewise, it is strongly recommended to organize several outdoor activities by creating a learning scenario that simulates the interaction between students and the natural environment. There is a consensus in recognizing the educational potential of playing and exercising in nature, and this value is especially interesting for schoolchildren with attention problems, since activities in nature favor self-control and concentration [79].

It has already been commented that one of the qualities that shows a clearer relationship between academic performance and executive functions, such as attention, is aerobic endurance. Therefore, its presence in physical education lessons with moderate intensities contributes to the improvement of fitness of students with ADHD [20]. Thus, creating running, bicycling, or skating clubs, in which the objective is to achieve different levels of difficulty, could obtain a great motivational environment [80–82]. In these activities, schoolchildren both individually, or in groups, can use smartwatches and cell phones to see their progression. Likewise, PE teachers can also know how his/her students´ progressions are taking place to give them the most appropriate feedback. When the students are aware of the number of meters, kilometers, and steps taken by comparing themselves, it could be an opportunity to promote the development of self-regulation among students, regardless of having ADHD or TD. Therefore, physical education teachers should ensure that schoolchildren, especially those with attention difficulties, do 20 to 30 min of moderate aerobic exercise in their classes, and they should promote it outside of school as well [17,22].

There is a huge number of studies that show low motor competence levels among students with ADHD [20]. Likewise, the majority of students with ADHD do not know anything about their motor actions or about how to carry them out together with coordination.

Therefore, they are not usually so motivated during the PE lessons [27]. Thus, it is imperative that, in physical education classes, the fundamental motor skills are strengthened and exercised both in a generic and specific way through all kinds of playful and sporting situations [80]. Furthermore, PE teachers should have a wide range of balancing and coordinating tasks, as well as agility tasks, such as gymnastics, which can be a challenge for their students at different educational levels, progressively increasing their difficulty and challenge, which would constitute a favorable vestibulo-cerebellar stimulus for all students and particularly for those with ADHD [80].

Including selected yoga, tai chi, or mindfulness activities in physical education classes can be a good decision that would benefit all schoolchildren, particularly those with attention and hyperactivity problems [80–82].

Physical education teachers can take advantage of the existence of tutorials and videos on these activities to transfer to their students those techniques that will be part of the class as a way of making them participate in it and as a way of offering them some advice on what is appropriate or not when selecting content from the network [81,82].

Finally, many schoolchildren, although they could participate in individual sports, would prefer to be able to practice team sports. If we consider that team sports are a powerful catalyst for social relations, as teachers we should encourage students with ADHD to participate in these kinds of sports. Likewise, PE teachers should promote, through team sports, that students with ADHD can make more friends. The thoughts and the traditional ways of developing sports may not work to include students with ADHD in PE lessons. In the same way, this it does not work with students with low motor skills or developmental coordination problems [80–82]. Listening to what students with ADHD think about the PE activities, the changes that they would make to the PE lessons, and how they would prefer to perform them can be good decisions to favor their inclusion where exclusion has been the norm. A PE teacher's mind must always be available to listen to the voice of the schoolchildren who present some type of difficulty in order to offer physical education classes in which everyone can enjoy their benefits [82].

## 5. Conclusions

It is important to highlight the role that the PE teacher and his teaching style can have in favoring the social exclusion of students with ADHD. Therefore, ways of teaching that facilitate participation, commitment to learning, and interaction among all students in the PE class must be taken into account. Likewise, it is important that PE teachers know the effects of physical exercise and sports on the nuclear symptoms of students with ADHD in order to adapt the motor tasks to the ecological circumstances and environment of PE class.

Finally, PE teachers have to become "students" of their students with ADHD to develop an "objective diagnosis" and always "prescribe" the best possible teaching.

### 5.1. Limitations

There was a wide range of studies under clinical and/or experimental conditions. However, data were limited in terms of ecological conditions. In addition, some involved young adolescents, whereas all the other studies involved children.

### 5.2. Future Prospects

Currently, facing a heterogeneous class can raise many doubts or difficulties in teaching, since it requires having to meticulously adjust and transform the entire didactic methodology following the principles of inclusion, diversification, and "impartial equity". To respond adequately, teachers and educators will have to be able to create "a school for each student". In this sense, physical education will have to be closer to science in order to adapt its findings related to ADHD and physical exercise, sports, and physical activity to their particular circumstances.

**Author Contributions:** Conceptualization, M.V.-d.G. and L.M.R.-P.; writing-original draft preparation, M.V.-d.G. and M.P.-N.; writing-review and editing, M.V.-d.G. and M.Á.G.-R. All authors have read and agreed to the published version of the manuscript.

**Funding:** This research received no external funding.

**Institutional Review Board Statement:** Not applicable.

**Informed Consent Statement:** Not applicable.

**Data Availability Statement:** Not applicable.

**Acknowledgments:** The authors are grateful to anonymous editors and reviewers for providing valuable insights on the discussed topics.

**Conflicts of Interest:** The authors declare no conflict of interest.

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
