# Peer review of "Attentional Neurodiversity in Physical Education Lessons: A Sustainable and Inclusive Challenge for Teachers"

_sustainability, doi:10.3390/su15065603_

Round 1

Reviewer 1 Report

The article a contingent problem for Physical Education teachers in relation to the attention to diversity and the perspective of educational inclusion. By  way of comments it is suggested:

-Revise title and abstract and incorporate TDHA as a construct on which the study is based.

-To make the theoretical and epistemological distinction betwen the integrating school and inclusive school.

-In the text, comments and observations of a prescriptive and normative nature are presented a more propositive language is suggested, wich expands the possibilities of intervention..

-To justify why university training should offer more qualification in the training of physical education teachers in terms of attention to diversity, especially children with TDHA, since in the perspective of inclusion it is an aspect of a wider renge that considers: gender, race, culture, religion, SEN , etc. 

-To strengthen the conclusions by virtue of the purpose and objective of the article. 

Author Response

Response 1. Thank you very much for your comments. We do not consider it necessary to incorporate the term and/or the acronym of ADHD in the title because “Attentional Neurodiversity” is including Attention Deficit/Hyperactivity Disorder indeed. Likewise, you can find this inclusion in the abstract of the manuscript.

Response 2.  We have included the following paragraph: "Likewise, inclusive education should be part of the integrating school where the teaching process involves all the children regardless of their educational needs."

Response 4. Between lines 415 and 423, we justify why the Spanish University should change in terms of training for future PE teachers. We show you the paragraph which was included to:

“However, it is well-known that the Spanish University does not provide enough specific knowledge to future PE teachers. Thus, they have to get a Master's Degree to acquire the knowledge that let them become qualified among the diversity in the mainstream schools”. In this sense, professors should provide students with education according to the needs of current society to form capable future teachers. This implies that apart from the knowledge of the professors, the university curriculum should be reviewed and changed to be oriented toward sustainability. Thus, it should be based on the theoretical and practical bases of inclusive education to be able to serve the characteristics of diversity present in classrooms (gender, capacity, culture, etc.)”

Response 5. We have rewritten the conclusions and we have added a future prospects:

Conclusions

A physical education class is supposed to be a context in which schoolchildren can participate and have fun learning the tasks set, improve their social skills, and interact abundantly with their peers, regardless of their level of development and difficulties. Thus, it is important to highlight the role that the PE teacher and his teaching style can have in favoring the social exclusion of students with ADHD, since many times the way of organizing the tasks, presenting them, or organizing them in the classes can cause these schoolchildren to interact less with their peers, and therefore, the number of opportunities to participate in class decrease, causing the possibility of being ignored or excluded to increase. Hence, ways of teaching that facilitate participation, commitment to learning, and interaction among all students in the PE class must be taken into account. Likewise, it is important the PE teachers know the effects of physical exercise and sports on the nuclear symptoms of students with ADHD in order to adapt the motor tasks to the ecological circumstances and environment of PE class.

Finally, PE teachers have to become “students” of their students with ADHD to develop an “objective diagnosis” and always “prescribe” the best possible teaching

Future prospects

Currently, facing a heterogeneous class can raise many doubts or difficulties in teaching, since it requires having to meticulously adjust and transform the entire didactic methodology following the principles of inclusion, diversification, and "impartial equity". To respond adequately, we will have to be able to create "a school for each student" without forgetting that many education professionals will need adequate training adapted to the new educational needs of the 21st century. In this sense, Physical Education will have to be closer to science in order to adapt its findings related to ADHD and physical exercise, sports, and physical activity, to their particular circumstances.

Reviewer 2 Report

The manuscript has a theoretical character and it aims to show the effectiveness of sports and physical exercise on the core symptoms of schoolchildren and adolescents with attention deficit/hyperactivity disorder. It has got a character of essay. Authors divided text into coherent subchapters and chapters. The text is written well chosen language and it is written in understandable form.

I have got only two small problems. The text includes many abbreviations, so they should be explained in some separated chapter.

The second one is regarding to references. Authors should eliminate references, which are not in English.

I hope my comments are helpful

Author Response

Response 1. Thank you for your comments. We have included in the manuscript the meaning of every acronym. 

Response 2. When the Spanish references were consulted, their titles did not appear in English (in square brackets) and this is the reason why we have included them only in Spanish.

Reviewer 3 Report

Thank you for submitting your valuable manuscript to this journal.

The aim of the present essay is to show the effectiveness of sports and physical exercise on the core symptoms of schoolchildren and adolescents with attention deficit/hyperactivity disorder. The authors wanted to describe the key differences between clinical effects of physical exercise and/or sports intervention and physical education lessons on students with attention problems. They emphasize the need of bridging the gap between Physical Education (PE) and science according to Agenda 2030.

There are some major points that need to be considered in the manuscript.

In general, I suggest you write different parts of the paper more concisely. The introduction part is too long, but without any explanation about attentional neurodiversity or attention deficit/hyperactivity disorder.

Section 2 of the paper is about Attention deficit/hyperactivity disorder (ADHD) definition and subtypes and lengthy explanations about ADHD students’ behavior which is unnecessary for this essay. It is needed to be summarized. Instead of these long explanations about the ADHD disorder, I would like to read the effects of PE lessons on the students with this disorder from the original papers. The whole manuscript is faced with a lack of original papers to validate the topics discussed.

Despite it is mentioned that there is a huge number of scientific studies that report the effectiveness of exercise and/or sports in the treatment of ADHD in section 3, again there is not enough evidence from the original papers here.

 Section 4 is also too long with lots of explanations without a clear and obvious subtitle. There is a lot of data without classification. You can classify each subject under a distinct subtitle and with a maximum of 4 to 5 paragraphs. Each paragraph should not be more that 5 to 8 lines.

Conclusion of the paper is too long. You have to summarize the conclusion and write it in a maximum of 2 paragraphs.

Good luck.

Author Response

Response 1. The reason why the introduction is long is to contextualize and adapt the manuscript to the topic of the Special Issue. Notwithstanding, we have included some information about ADHD and Attentional Neurodiversity as part of the purposes of the manuscript at the end of the introduction: "Because of the growing presence of ADHD among children and adolescents in schools, as the main part of attentional neurodiversity, this essay aims to highlight how important is to bridge a gap between science and PE to respond to the educational needs of them." 

Response 2. We have considered the need of including the psychosocial characteristics of students with ADHD to help the readers for understanding the effects of Physical Education on students with ADHD.

Response 3. Although all the information is duly cited and based on the most relevant and current scientific literature, the truth is that there is not enough research carried out under ecological conditions; therefore, there is little research on the effects of physical education lessons on the psychosocial aspects of students with ADHD. However, we have included the main investigations according to the effects of PE classes on the psychosocial factors of students with ADHD.

Response 4. Section 3 shows the most important research related to the benefits of physical exercise and/or sports on the nuclear symptoms of ADHD. As we can see, the majority of studies included were developed under clinical or experimental conditions that are so far away from the real PE lessons conditions (ecological conditions). Therefore, section 3 could well open the way to the showing of section 4 which talks about the distance between science and PE.

Response 5. We have included some subtitles (from 4.1. to 4.7) to clarify the structure and content of section 4. Otherwise, we have only considered making smaller the following subtitles: 4.1., 4.2., 4.3. The rest of the subtitles included in the manuscript consist of the same content because show quite relevant scientific literature that can provide PE teachers with good tools to respond to the educational needs of students with ADHD (please see the attachment).

Response 6. We have rewritten the conclusions and we have added future prospects:

Conclusions

A physical education class is supposed to be a context in which schoolchildren can participate and have fun learning the tasks set, improve their social skills, and interact abundantly with their peers, regardless of their level of development and difficulties. Thus, it is important to highlight the role that the PE teacher and his teaching style can have in favoring the social exclusion of students with ADHD, since many times the way of organizing the tasks, presenting them, or organizing them in the classes can cause these schoolchildren to interact less with their peers, and therefore, the number of opportunities to participate in class decrease, causing the possibility of being ignored or excluded to increase. Hence, ways of teaching that facilitate participation, commitment to learning, and interaction among all students in the PE class must be taken into account. Likewise, PE teachers must know the effects of physical exercise and sports on the nuclear symptoms of students with ADHD to adapt the motor tasks to the ecological circumstances and environment of PE class.

Finally, PE teachers have to become “students” of their students with ADHD to develop an “objective diagnosis” and always “prescribe” the best possible teaching

Future prospects

Currently, facing a heterogeneous class can raise many doubts or difficulties in teaching, since it requires having to meticulously adjust and transform the entire didactic methodology following the principles of inclusion, diversification, and "impartial equity". To respond adequately, we will have to be able to create "a school for each student" without forgetting that many education professionals will need adequate training adapted to the new educational needs of the 21st century. In this sense, Physical Education will have to be closer to science to adapt its findings related to ADHD and physical exercise, sports, and physical activity, to their particular circumstances.

Reviewer 4 Report

Dear author,

The aim of this essay aims to show the effectiveness of sports and physical exercise on the core symptoms of schoolchildren and adolescents with attention deficit/hyperactivity disorder. The manuscript has a current theme and some potential, but the study design should be improved.

The manuscript structure is actually a book chapter structure rather than an essay. I recommend that it be transformed into a narrative review. Also, a section should be added on the strategies, methodologies and methods with the most validity in the literature for the research topic under review.

Additionally, the conclusions should be written in a shortly and objectively. Also, add a section with criticisms/limitations, practical applications and future prospects for Attentional Neurodiversity.

Good work!

Author Response

Response 1. An analysis of the literature within the field of study is offered, identifying possible gaps and/or existing problems. In addition, the reader is continually invited to rethink the subject matter. Therefore, attention is paid to the format and nature of a scientific essay instead of a narrative review.

Response 2. We have rewritten the conclusions and we have added future prospects:

Conclusions

A physical education class is supposed to be a context in which schoolchildren can participate and have fun learning the tasks set, improve their social skills, and interact abundantly with their peers, regardless of their level of development and difficulties. Thus, it is important to highlight the role that the PE teacher and his teaching style can have in favoring the social exclusion of students with ADHD, since many times the way of organizing the tasks, presenting them, or organizing them in the classes can cause these schoolchildren to interact less with their peers, and therefore, the number of opportunities to participate in class decrease, causing the possibility of being ignored or excluded to increase. Hence, ways of teaching that facilitate participation, commitment to learning, and interaction among all students in the PE class must be taken into account. Likewise, the PE teachers must know the effects of physical exercise and sports on the nuclear symptoms of students with ADHD to adapt the motor tasks to the ecological circumstances and environment of PE class.

Finally, PE teachers have to become “students” of their students with ADHD to develop an “objective diagnosis” and always “prescribe” the best possible teaching

Future prospects

Currently, facing a heterogeneous class can raise many doubts or difficulties in teaching, since it requires having to meticulously adjust and transform the entire didactic methodology following the principles of inclusion, diversification, and "impartial equity". To respond adequately, we will have to be able to create "a school for each student" without forgetting that many education professionals will need adequate training adapted to the new educational needs of the 21st century. In this sense, Physical Education will have to be closer to science to adapt its findings related to ADHD and physical exercise, sports, and physical activity, to their particular circumstances.

Round 2

Reviewer 3 Report

Thank you for the revision of your valuable manuscript.

The paper is still too long and it looks like a book chapter rather than an essay. Very few original papers have been used in this manuscript, but it emphasizes the importance of physical exercise and sports as educational needs of ADHD schoolchildren.

I have no more comments than before.

Good luck.

Author Response

Thank you for your comments. The lack of original papers is because most studies were under clinical conditions. Notwithstanding, we have included it as a limitation of the essay. Likewise, we have included some changes to improve the manuscript (please see the updated version).

Yours faithfully,

Reviewer 4 Report

Dear authors,

Thank you very much for sending the second revised version. It is noted that the authors have corrected the first version, however I leave some minor revisions to increase the robustness of the manuscript. 

Please see cover letter in attachment.
